# Minimal Information for Studies of Extracellular Vesicles (MISEV): Ten-Year Evolution (2014–2023)

**DOI:** 10.3390/pharmaceutics16111394

**Published:** 2024-10-29

**Authors:** Yuan Zhang, Mengyi Lan, Yong Chen

**Affiliations:** 1School of Pharmacy, Jiangxi Medical College, Nanchang University, Nanchang 330006, China; yuanzz2000@163.com (Y.Z.); lanmengyi0811@163.com (M.L.); 2Jiangxi Key Laboratory for Microscale Interdisciplinary Study, Institute for Advanced Study, Nanchang University, Nanchang 330031, China

**Keywords:** extracellular vesicles (EVs), international society for extracellular vesicles (ISEV), minimal information for studies of extracellular vesicles (MISEV), exosomes, ectosomes, microvesicles, exomeres, supermeres, machine learning

## Abstract

In the tenth year since the first edition of MISEV was released in 2014, MISEV2023 has been reported in 2024 with the aim of refining the standard and improving the rigor, reproducibility, and transparency of extracellular vesicle (EV) research to clarify the requirements for experimental design of EVs, emphasize the importance of reproducible experimental results as well as encouraging openness of experimental information. The release of MISEV has significantly contributed to the quality of research in the field of EVs, which creates a more reliable research environment. However, despite the important role of MISEV, there is still a need for the EV researchers to continue to push for the widespread implementation of the guidelines to meet the evolving nature and challenges of EV research. The evolution of EV research and the attention it receives have grown exponentially over time, as has the number of people involved in the writing of MISEV. Here, this review briefly summarizes the evolution of the three editions of MISEV, aiming to recall MISEV2014 and MISEV2018 while learning about the latest release, MISEV2023, to gain a deeper understanding of the content, and to provide a quick note for beginners who want to learn about MISEV and explore the EV world.

## 1. Brief History of EVs, ISEV, and MISEV

Extracellular vesicles (EVs), as particles released from cells with a lipid bilayer structure, are of interest because of their important role in intercellular communication, signaling, and immunomodulation. The first discovery of EVs began with a coagulation study in the 1940s, when Erwin Chargaff and Randolph West found that blood precipitates that had been centrifuged at high speeds had significant coagulation properties which are now recognized as extracellular vesicles [1,2]. To provide guidance on the best methodological practices in the field of EV research, the International Society for Extracellular Vesicles (ISEV) was created in 2011. What followed was the creation of Minimal Information for Studies of Extracellular Vesicles in 2014 (MISEV2014) by Lötvall et al. [3], which provided researchers with a minimum set of biochemical, biophysical, and functional criteria. However, as EVs received more attention, MISEV2014 was no longer able to meet the current needs of the EV field, and so relevant researchers, e.g., Clotilde Théry et al., revamped and upgraded MISEV2014, with the birth of MISEV2018 [4]. In the 10th year of the publication of MISEV2014, MISEV2023 authored by Welsh et al. aims to refine the standards and improve the rigor, reproducibility, and transparency of EV research [5,6]. By providing clear definitions and classifications, detailed guidelines for experimental design, data reporting requirements, and standardization of sample treatment, MISEV2023 ensures the reliability and reproducibility of EV research. MISEV2023 encourages the sharing of experimental information and promotes the construction of databases to enhance the openness and transparency of studies. This guideline builds a solid foundation for basic research in the field of EVs. On the occasion of the 10th anniversary of the publication of MISEV, this review summarizes and compares the three versions of MISEV in order to help EV researchers have a deeper understanding of MISEV and of EV development/changes from the perspective of MISEV.

## 2. Analysis, Generalization and Comparison of Three MISEV Versions

Here, this review briefly summarizes the main contents of the three versions of MISEV so that readers can have an initial understanding of the guide, in addition to a more visual comparison of the contents. This review focuses on the main changes in seven aspects of MISEV in the ten years.

### 2.1. Nomenclature

Figure 1 shows the recommended and discouraged extracellular particle (EP) nomenclature in MISEV2023. A comparative display of MISEV terminology is shown in Table 1. Regarding the definition of EVs, it can be found that with the update of MISEV, the 2018 edition is the most stringent in defining EVs, as the 2023 edition deleted the term “naturally” to avoid inadvertent exclusion of engineered EVs or EVs produced in a variety of cell culture conditions. In terms of nomenclature, MISEV2023’s recommended EV nomenclature is basically the same as the previous version, and it also emphasizes the “non-vesicular extracellular particles” (NVEPs), such as “lipoproteins”, “exomeres” and “supermeres” [7,8], that are enriched in most EV formulations. Therefore, it is important to provide more detailed recommended guidance on the nomenclature of EV subtypes and NVEPs in order to avoid misuse of biogenesis terms such as “exosomes”, “ectosomes” and “microvesicles”. 

### 2.2. Collection and Pre-Processing: Pre-Analytical Variables Through to Storage

The chapter of Collection and pre-processing has been covered since MISEV2018, which addressed the fact that a range of factors in the collection, pre-processing, and storage of samples containing EVs and the derivatives can affect EVs both quantitatively and qualitatively, making standardization of manipulation and reporting of variables a primary step. MISEV2023 builds on this foundation by adding blood, milk, and urine-derived sources, which were not discussed in detail in the previous editions, as well as new descriptions of bacterial, cerebrospinal fluid, and salivary-derived sources (Figure 2). Depending on the source of the sample (e.g., cell culture medium (CCM), blood, urine, etc.), there are significant differences in the methods used to collect samples and to isolate and store EVs. Therefore, a detailed documentation of the entire experimental conditions and procedures is essential to ensure the reproducibility of the experiments and the reliability of the results. For CCM, the report should include, but not be limited to, the composition and preparation of the medium, characteristics of EV-producing cells, culture conditions, frequency and method of cell culture medium harvesting, CCM storage, and whether serum, platelet lysate, or other complex additives were used. For blood samples, it is necessary to record donor characteristics, comply with blood collection requirements, avoid/remove platelet activation, and report that EVs are enriched in blood samples. With the support of MIblood-EV, researchers can significantly improve the quality and reproducibility of studies involving blood EVs [9]. During urine processing, care needs to be taken to use cell-free urine and to document the method and results of the EV isolation considered, and the data on urinary EV and non-EV urine parameters should be collected. For more details, you can refer to the guidelines for urine EVs [10,11]. When isolating and storing EVs, it is important to regulate environmental conditions (e.g., pH and temperature) to maintain their functionality. Therefore, comprehensive and detailed experimental records not only help data analysis, but also provide the basis and reference for subsequent studies.

Due to the wide range of EV sources, with over 400 cell types in the human body [12], for example, it is not enough to describe the above sources. In order to include other sources of EVs more comprehensively, ISEV has established a Task Force to keep researchers updated and aware of the results of the Task Force (https://www.isev.org/taskforces, accessed on 26 October 2024). As for EV storage, due to the diversity of EVs, there continues to be a lack of harmonized recommendations or standards for EV storage conditions. Therefore, storage conditions, including any additives, should be adequately reported and the impact on EV quantity and quality should be studied. Table 2 displays a comparison of sample collection and pre-processing among three editions of MISEV.

### 2.3. EV Separation and Concentration

This chapter begins with MISEV2014, which required the reporting of all details on reproducible methods for isolating EVs. Then, MISEV2018, based on MISEV2014, evaluated the specificity and recovery of the methods of isolation and added the evaluation of commercial kits. Now, in MISEV2023, traditional separation methods are further refined and additional separation and concentration means, such as fluid flow-based separation techniques, are added. A brief description of each method and reporting recommendations in MISEV2023 are detailed in Table 3.

Comparing the three editions of MISEV shows that there is no one-size-fits-all approach, as the choice of separation and concentration methods must vary according to factors in different studies. In the process of separating EVs, EV preparations are often accompanied by other components. For example, LDL is more similar to exosomes due to its shape and size; density gradient centrifugation, size exclusion chromatography (SEC), and ultrafiltration (UF) are more suitable to separate the two. And for soluble proteins, the use of immunocapture (IC) is more appropriate [13]. Of course, the combination of separation techniques is more helpful to achieve high-purity samples, and both Density gradient ultracentrifugation (DGUC)-SEC and SEC-fast protein liquid chromatography (FPLC) combinations have significantly reduced EV/lipoprotein contamination [14,15]. In addition to this, methods such as utilizing styrene-maleic acid [16], the Simoa assay for ApoB-100 [17], and magnetic bead reagents [18] can be used to improve extracellular vesicle purification. For EV separation from cell culture media and blood, samples can be separated by combining SEC with UF to improve purity and yield [19,20], and even a three-step protocol of polyethylene glycol (PEG) precipitation, gradient centrifugation, and SEC can be utilized for efficient removal of impurities and recovery of EVs [21]. Automated separation and hierarchical screening of size-based subpopulations have also been applied in the online immunoaffinity chromatography–asymmetric flow field flow separation (IAC–AF4) technique [22]. In conclusion, the selection of suitable separation strategies is crucial for different sample types, which will significantly affect the efficiency of EV extraction and the success of subsequent analysis. With the advancement of technology, emerging methods and combinations continue to emerge, providing more possibilities for efficient EV purification.

EV-TRACK has been recommended since MISEV2018 as an online database for transparent reporting in EV research. Its purpose is to standardize EV research, facilitate the application of experimental protocols, and enable better interpretation and reproducibility of experimental results with the aim of enhancing the rigor, reproducibility, and transparency of EV experiments. It contains methodological parameters of EV-related literature, based on MISEV to assess the parameters related to the isolation and identification of EVs. As of 25 October 2024, the current data in this database include 3413 publications and 9748 experiments with continuous updating [23].

Throughout MISEV2023, it can be seen that although the separation and purification techniques for EVs as a whole have not changed significantly, the continuous standardization of operating procedures and the deepening of technical exchanges, the effective combination of different methods, and the continuous invention and validation of numerous new technologies have brought us closer to the goal of realizing the efficient purification of EV preparations.

### 2.4. EV Characterization

MISEV2014 provided general recommendations for characterizing proteins with “three positives and one negative”, whereas for the characterization of individual vesicles, two different but complementary techniques need to be used. However, the guideline does not give quantitative recommendations, which are supplemented by MISEV2018, mentioning that both the source of EVs and the preparation of EVs must be quantitatively characterized. MISEV2018 builds on the “three positives and one negative” recommendations for protein characterization by proposing a five-component framework for EV protein characterization, which divides proteins into five categories for researchers to choose the proteins to characterize. MISEV2023 follows and updates the five-component framework. Proteins like CD9, CD63, CD81, TSG101, ALIX, and GAPDH can be used to assess the presence of EVs, but it is worth noting that not all EVs display these proteins. Notably, in addition to testing for EV marker proteins, the presence of NVEP in EV preparations needs to be evaluated to ensure sample purity and specificity. Regarding the characterization of individual vesicles, MISEV2018 and MISEV2023 further complement the current characterization techniques.

Notably, MISEV2018 adds a recommendation on the topological relationship between specific proteins and EVs, noting that it is necessary to experimentally demonstrate the spatial relationship of some proteins and nucleic acids associated with EVs. MISEV2023 adds the need to provide an indication of the instrument/method limit of detection (LOD) when using quantitative metrics to characterize EVs. MISEV2023 discusses different methods of EV characterization in separate sections, each of which provides recommendations (Table 4).

To ensure reliable and reproducible interpretation of data, MISEV2023 lists commercially available and literature-supported techniques and details instrument-specific reporting considerations, which can be viewed as complementary to EV characterization. It is intended to standardize the rigorous use of characterization techniques more comprehensively to better refine our experimental methods. Techniques like flow cytometry [24] and mass spectrometry proteomics [25] have published their standards or report recommendations.

New advances in high-resolution microscopy techniques, in conjunction with innovative labeling strategies, have made it possible to explore kinetics and pharmacogenetics at the nanoscale [26,27,28,29]. The application of total internal reflection fluorescence microscopy (TIRFM) with dynamic correlative light and electron microscopy (CLEM) provides a powerful tool for observing the fusion of EVs with the cytoplasmic membrane and real-time in vivo imaging of EVs [30,31,32]. By utilizing TIRFM visualization for automatic image acquisition and quantification to design a tunable micropattern-array assay detection method in realizing EV sorting while detecting its RNA and proteins in situ, it simplifies the operation process and improves the measurement accuracy [33]. Single-particle interferometric reflectance imaging sensing (SP-IRIS) technology in combination with single-molecule fluorescent in situ hybridization (smFISH) can likewise detect the RNA and surface proteins of EVs. In terms of characterizing the size of EVs, in addition to the widely used nanoparticle tracking analysis (NTA), dynamic light scattering (DLS), and resistive pulse sensing (RPS), nuclear magnetic resonance (NMR) has gradually emerged as an effective means that can be used to measure the size of extracellular vesicles [34]. Limited by factors such as EV heterogeneity and yield, studies have begun to use only flow cytometry instead of WB or both flow cytometry and WB techniques as a means of detecting EV markers [35,36].

### 2.5. EV Release and Uptake

This chapter is also unique in MISEV2023, but can be compared to the subsection “Determining whether exosomes are functionally specific compared to other small EVs” in MISEV2018, which discusses inhibition of EV release through a range of genetic manipulations and drugs, but does not give recommendations. Understanding and being able to regulate EV release and uptake will help researchers further explore EV functions and will be important for demonstrating, for example, the strength of the association of specific EV subtypes with specific functions, lipophilic dyes leading to false positive signals, and the impact of EV uptake pathways on specific functions. Therefore, this content in MISEV2023 discusses and makes recommendations on how to observe or modulate EV release and uptake.

Learning about the uptake and release mechanisms of EVs can deepen our understanding of intercellular communication and provide new perspectives on the application of EVs as drug carriers. EVs possess the ability to target specific cells, good biocompatibility, and excellent drug-carrying capacity, which enables them to efficiently encapsulate small-molecule drugs, RNAs, and proteins. In addition, EVs are able to penetrate cell membranes and release drugs directly into target cells, while protecting them from degradation, thus improving bioavailability [37,38,39,40]. These properties have enabled EVs to show a wide range of potential applications in therapeutic areas such as cancer, cardiovascular diseases, autoimmune and neurological diseases [41,42].

### 2.6. Functional Studies

MISEV2014 mentions the effects of dose gradients, negative controls, high-purity EV and non-EV components on function to demonstrate, as far as possible, that EV is inextricably linked to functional activity, and to exclude false positives from medium and non-EV components of the functional analysis. MISEV2018 continues to follow these recommendations and adds the requirement for quantitative comparisons as well as confirmation of the percentage of EV activity. In turn, MISEV2023 adds further details to improve the corresponding content (Table 5). When conducting functional studies, dose-response and time-course assays are recommended for a comprehensive assessment of the biological effects of EVs. Appropriate negative EV controls should be used to ensure the accuracy of the experiments. In addition, non-EV negative controls need to be evaluated to exclude possible background effects. This will help improve the reliability and relevance of the findings, leading to a better understanding of EV functions and mechanisms.

### 2.7. EV Analysis In Vivo

MISEV2023 summarizes in vivo EV studies for the first time, aiming to improve the understanding of the diversity of in vivo research. It lists several in vivo models (e.g., budding yeast, green algae, nematode, etc.) and summarizes the research advantages, the genetic susceptibility, and genetic similarity to humans of the corresponding in vivo models. When performing in vivo experiments, detailed reporting of labeling, detection techniques and exogenous EV administration parameters is required to facilitate replication studies. Also, attention is paid to the effects of EV labeling on biodistribution and function, as well as behavioral differences between endogenous and exogenous EVs. In vivo EV experiments are of great significance in the fields of cell biology, disease research, and drug development, and can provide new ideas and methods for scientific research and medical applications.

## 3. Summary and Perspectives

Over the past ten years, from 2014 to 2023, the number of individuals involved in the development of MISEV has grown exponentially, expanding from dozens to hundreds, and ultimately surpassing a thousand contributors shaping MISEV2023 [43]. This growth signifies an increasing interest and commitment to EV research. The continuous updates to MISEV aim to enhance the visibility of EVs and ensure that the field develops in a healthy manner. The evolution of MISEV can be likened to a flourishing tree, providing shade and support for researchers engaged in EV studies. MISEV2023 builds upon the foundations laid by MISEV2014 and MISEV2018, offering more concise recommendations and refining previous concepts while incorporating the latest technologies. It encompasses commonly used and advanced techniques, introducing new sections on technique-specific reporting considerations for EV characterization, release and uptake, and in vivo analysis.

MISEV has been updated not only in terms of technology, but also in terms of researchers’ deeper understanding of EV characteristics and applications. In the future, these innovations and collaborations will drive the development of more reproducible, transparent, and rigorous experimental methods, making the process of studying EV easier and more efficient, and thus providing higher-quality studies for both basic research and clinical applications. Although MISEV2023 provides important guidance for EV research, it still faces some limitations and challenges. The lack of transparency of the experimental process makes it difficult to repeat experiments, which is also highlighted by MISEV2023. Second, EV isolation techniques in complex samples need to be improved, and there is a lack of efficient and simple isolation techniques. And some techniques still lack uniform standards. The functional mechanisms of EVs are still unclear, so there is still a long way to go to apply EVs to clinical use. In addition, it is worth noting that MISEV2023 lacks the categorization to distinguish some terms like cell membrane-derived vesicles [44,45], which are similar to EV characteristics, to avoid conceptual confusion among researchers. Terms such as cell membrane-bound vesicles are defined to correspond to artificial cell-derived vesicles (ACDVs) in MISEV2023 [46,47]. However, MISEV2023 lacks the addition of similar terms. It is helpful for readers to clearly and quickly distinguish EV related terms by the addition of similarly defined terms.

In the current thriving landscape of machine learning, the EV field is also increasingly adopting machine learning techniques. These techniques are being used for EV recognition, classification, and component detection, greatly advancing the progress of disease diagnosis and prediction. For example, machine learning was used to identify, classify, and quantify EVs, and the purity and inflammatory state of EVs were assessed through unsupervised machine learning [48]. This unsupervised learning method is able to automatically analyze data structures without pre-labeling, helping researchers better understand how EVs behave in different physiological and pathological states. Machine learning can also be used to analyze the components of EVs such as proteins and RNAs. Through the combination of deep learning and TIRF, miRNAs of EVs were detected while their cancer origin was analyzed and diagnosed [49]. Not only that, but the results of EV proteomics analysis with the help of machine learning are used to predict tumor invasion and proliferation ability and identify EVs of cancer origin, which is of great significance for revealing disease signaling pathways and biological processes and identifying potential drug targets [50,51,52,53]. Overall, the introduction of machine learning has brought new opportunities for EV research. Its application not only improves the efficiency and accuracy of experiments, but also provides new perspectives for clinical practice and promotes the development of precision medicine. In the future, whether the next version of MISEV will incorporate recommendations on machine learning is indeed a topic worthy of exploring in depth.

While updating MISEV is a lengthy process, the rapid advancements in EV research necessitate timely recording, reporting, and sharing of experimental processes and findings, as advocated by MISEV2023. This practice contributes to more robust and comprehensive EV research and informs future iterations of MISEV. To foster universal participation, ISEV recommends utilizing Task Force and EV-TRACK for researchers to upload and exchange their experimental data. This approach creates a collaborative environment where data and suggestions can be dynamically updated in real time, potentially leading to a scenario where every second represents a new version of MISEV.

## Figures and Tables

**Figure 1 pharmaceutics-16-01394-f001:**
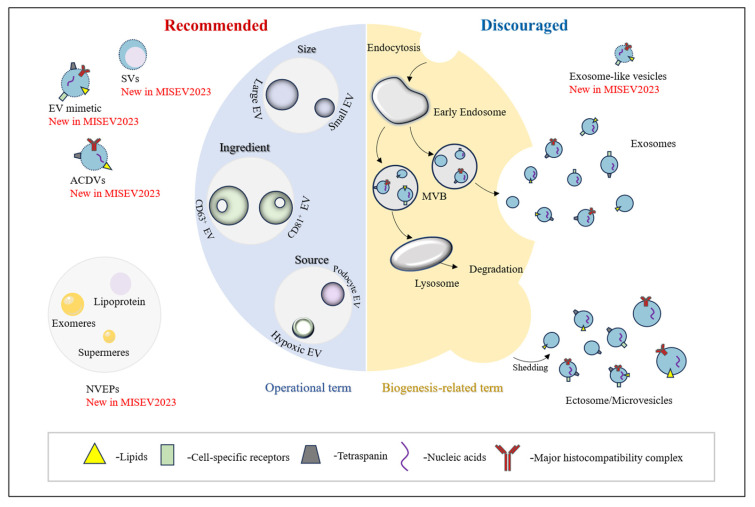
Recommended and discouraged extracellular particle (EP) nomenclature in MISEV2023. Left: Recommended EV nomenclature based on operational terms such as physical characteristics (e.g., small/large EVs), biochemical composition (e.g., CD63+/CD81+ EVs), cellular origin and/or conditions under which EVs are produced (e.g., podocyte, hypoxic EVs), and related recommended terms. Right: Discouraged EV nomenclature for terms related to the biogenesis, like exosomes, ectosomes and microvesicles, and related discouraged terms. Abbreviations: ACDVs, artificial cell-derived vesicles; MVB; multivesicular body; NVEPs, non-vesicular extracellular particles; SV, synthetic vesicles.

**Figure 2 pharmaceutics-16-01394-f002:**
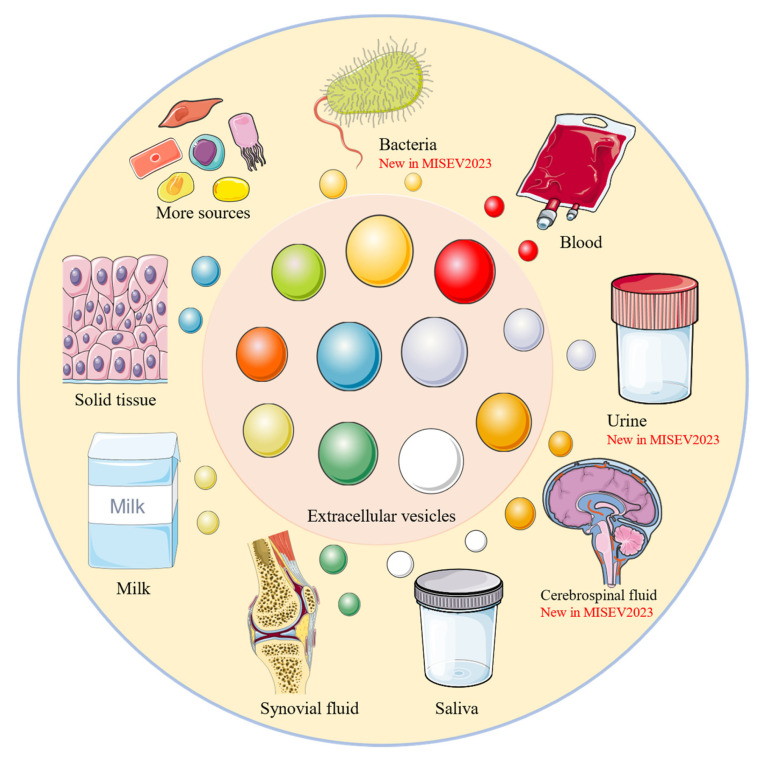
EV sources discussed in MISEV. Bacterial, cerebrospinal fluid, and salivary-derived sources are new descriptions in MISEV2023 compared with MISEV2018. Figure was partly generated using Servier Medical Art, provided by Servier, licensed under a Creative Commons Attribution 3.0 unported license.

**Table 1 pharmaceutics-16-01394-t001:** Comparison of nomenclature in MISEV.

Nomenclature	
**MISEV2014**	**Definition**: Extracellular vesicles (EVs) are the secreted membrane-enclosed vesicles. No recommendations for nomenclature.
**MISEV2018**	**Definition**: EVs are particles naturally released from cells that are defined by a lipid bilayer and cannot replicate, i.e., do not contain a functional nucleus. Unless the authors have been able to establish specific markers of subcellular origins that are reliable in their experiments, e.g., real-time imaging techniques that capture the process of EV release, authors are urged to consider the use of operational terms for EV subtypes that refer to (a) the physical characteristics of EVs; (b) the biological constituents; (c) the replacement of terms such as exosomes and microvesicles with descriptions of the conditions or cellular origins.
**MISEV2023**	**Definition**: EVs refer to particles that are released from cells, are delimited by a lipid bilayer, and cannot replicate on their own (i.e., do not contain a functional nucleus). Authors are recommended to use the generic term “EV” and operational extensions of the term (consistent with MISEV2018) rather than inconsistently defined and sometimes misleading terms such as “exosomes” and “extracellular bodies” associated with difficult-to-establish biogenesis pathways, unless such EV populations have been specifically isolated and characterized.

**Table 2 pharmaceutics-16-01394-t002:** Comparison of sample collection and pre-processing in MISEV.

Collection and Pre-Processing: Pre-Analytical Variables Through to Storage	
**MISEV2014**	-
**MISEV2018**	A range of factors (including the characteristics of the source, how the source material is handled and stored, and experimental conditions) may affect EV recovery. Therefore, it is critical to plan collection and experimental procedures to maximize the number of known, reportable parameters, and then to report as many known pre-analytical parameters as possible.
**MISEV2023**	Report in detail on a range of factors in sample collection, pretreatment, and storage of sources and their derivatives containing EVs that may affect EVs both quantitatively and qualitatively.

**Table 3 pharmaceutics-16-01394-t003:** Comparison of EV separation and concentration in MISEV.

EV Separation and Concentration	
**MISEV2014**	(a) There is no single optimal separation method, so choose based on the downstream applications and scientific question. (b) Report all details of the methods for reproducibility. **No mentioned techniques/methods.**
**MISEV2018**	(a) A categorized review of currently common separation means based on EV recovery and specificity. (b) All details of reproducible methods are reported. ISEV strongly recommends that authors deposit experimental details in EV-TRACK **(New in MISEV2018).** (c) Recommendations for commercial kits **(New in MISEV2018).** **Mentioned techniques/methods:** Differential ultracentrifugation; Density gradients; Precipitation; Filtration; Size exclusion chromatography (SEC); Immunoisolation; Affinity isolation; Tangential flow filtration (TFF); Field-flow fractionation (FFF); Asymmetric flow field-flow fractionation (AF4); Ion exchange chromatography; Deterministic lateral displacement (DLD) arrays; Field-free viscoelastic flow; Alternating current electrophoretics; Acoustics; Microfiltration; Fluorescence-activated sorting; Lipid affinity; Hydrostatic filtration dialysis; Fast protein/high perfomance liquid chromatography (FPLC/HPLC).
**MISEV2023**	The choice of any isolation method should be based on the known properties of the specific EV sources as well as the desired EV yield and specificity. **Mentioned techniques/methods:** Differential ultracentrifugation; Density gradients; Precipitation; Filter concentration; SEC; Immuno-precipitation (IP); Affinity precipitation (AP); TFF; FFF; AF4; Ion exchange chromatography; Free-flow electrophoresis (FFE); Commercial kits.

**Table 4 pharmaceutics-16-01394-t004:** Comparison of EV characterization in MISEV.

EV Characterization	
**MISEV2014**	(a) General characterization. i. At least three EV positive protein markers, including at least one transmembrane/lipid binding protein and cytoplasmic protein. ii. At least one negative protein marker. (b) Characterization of single vesicles: Use of two different but complementary techniques. (c) No quantitative recommendations. **Mentioned techniques/methods:** Western blots (WB); Flow cytometry (FACS); Global proteomic analysis; Electron microscopy; Atomic force microscopy (AFM); Transmission electron microscopy (TEM).
**MISEV2018**	(a) Both the source and the preparation of the EVs must be quantitatively characterized. (b) General characterization. Continue with the “three positives and one negative” and introduce a five-component framework **(New in MISEV2018).** (c) Characterization of single vesicles: continue as before, but add more techniques. (d) Determine the topology of components related to EVs **(New in MISEV2018).** **Mentioned techniques/methods:** Nanoparticle tracking analysis (NTA); FACS; WB; Electron microscopy; AFM; TEM; High-resolution FACS; Resistive pulse sensing (RPS); Cryo-EM; DLS; Sodium dodecyl sulfate polyacrylamide gel electrophoresis (SDS-PAGE); Fourier-transform infrared spectroscopy (FTIR); Capillary electrophoresis; Enzyme-linked immunosorbent assay (ELISA) bead-based FACS; Surface plasmon resonance (SPR); Infrared (IR) spectroscopy; Raman spectroscopy (RS); Fluorescence microscopy; Confocal microscopy; Scanning electron microscopy (SEM); Scanning-probe microscopy (SPM); Super-resolution microscopy; Multi-angle light scattering combined with asymmetric flow field-flow fractionation (AF4-MALS); Fluorescence correlation spectroscopy (FCS); Raman tweezers microscopy; Single-particle interferometric reflectance imaging sensing (SP-IRIS) **(New in MISEV2018).**
**MISEV2023**	(a) Each EV preparation should be defined by quantitative measures of the source of EVs. (b) Approximations of the abundance of EVs should be made. (c) EV preparations should be tested for the presence of components associated with EV subtypes or EVs generically, depending on desired specificity one wishes to achieve. (d) Establish the degree to which non-vesicular, co-isolated components are present. (e) Provide an indication of the instrument/method limit of detection (LOD) when EVs are characterized with quantitative metrics **(New in MISEV2023).** **Mentioned techniques/methods:** RPS; NTA; DLS; FACS; Multi-angle light scattering; cryo-EM; FTIR; Chromatography; Capillary electrophoresis; Isolation kits; Liquid chromatography; SEM; TEM; Cryo-EM; SPM; AFM; Lipid mass spectrometry; RS; Bead-based FACS; Mass spectrometry (MS); Confocal microscopy; WB; SP-IRIS; Genetic protein tagging; Agilent Bioanalyzer pico chip; NanoDrop; Qubit microRNA Assay kit; Triple-quadrupole (QQQ) liquid chromatography (LC)-MS; Total Internal Reflection Microscopy (TIRFM); Light-sheet microscopy; HPLC; Interference reflectance imaging sensor (IRIS); Fluorescent super-resolution microscopy; Quantitative polymerase chain reaction (qPCR); Digital PCR; Droplet digital PCR (ddPCR) **(New in MISEV2023).**

The blue fonts indicate the additional content compared to the previous version of MISEV.

**Table 5 pharmaceutics-16-01394-t005:** Comparison of functional studies in MISEV.

Functional Studies	
**MISEV2014**	(a) The dose–function relationship should be quantitatively analyzed when isolated EVs are used for in vitro functional studies. (b) Systemic negative controls should show minimal functional impact. (c) Assessing the impact of soluble or non-EV macromolecular components.
**MISEV2018**	(a) Quantitative comparison of activity in conditioned media or biofluids before EV elimination, after EV elimination, and in the EVs is needed, as well as quantitative comparison of activity against the target EV subtype versus the “discarded” EV subtype. (b) Set more stringent negative controls. (c) Functional assays are recommended after rigorous isolation to compare EV and non-EV fractions to determine the proportion of activity associated with each fraction.
**MISEV2023**	(a) Encourage physiologic dose-response and time-course studies. (b) EV negative controls need to be carefully selected to assess the contribution of “background” EV activity and/or non-specific activity other than the EVs of interest. (c) Controls consisting of EV-free, EV-depleted, or enzyme-treated EV isolation fractions can help determine whether a function is EV-specific or associated with co-isolated materials.

## Data Availability

All data generated or analyzed are included in this article.

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
