# Peer review of "Minimal Information for Studies of Extracellular Vesicles (MISEV): Ten-Year Evolution (2014–2023)"

_pharmaceutics, 2024, doi:10.3390/pharmaceutics16111394_

Round 1

Reviewer 1 Report

Comments and Suggestions for Authors

The manuscript entitled “Minimal information for studies of extracellular vesicles (MISEV): Ten-year evolution (2014-2023)” by Y Zhang et al. is clearly written, original, and well-organized.

The manuscript provides a comprehensive overview of the evolution of the "Minimal Information for Studies of Extracellular Vesicles" (MISEV) guidelines, detailing the progression and updates made in the 2014, 2018, and 2023 versions. The authors effectively capture the rationale behind each iteration, focusing on the evolving technological advancements and the challenges faced by the extracellular vesicle (EV) research community. This work is timely and valuable, as MISEV guidelines have become a cornerstone for standardized EV research, ensuring reproducibility and transparency in a rapidly growing field.

The manuscript does an excellent job of documenting the historical evolution of the MISEV guidelines. The authors present a well-structured narrative that not only emphasizes the motivations behind each version but also provides a broader context for why such guidelines are necessary for the EV community. A major strength of the paper is its detailed comparison between the different versions (2014, 2018, and 2023) of the guidelines. By summarizing the key changes in each iteration, the authors clearly demonstrate how the MISEV guidelines have evolved to accommodate new technologies and meet the challenges associated with reproducibility.

The manuscript is well-organized, with clear sections that help readers navigate through different aspects of the guidelines' evolution. Each version is thoroughly explained, providing insight into the considerations that led to changes.

Concerns

  1. Lack of Practical Examples: While the manuscript outlines the differences between the various versions of the MISEV guidelines, it would benefit from including specific practical examples of how these guidelines have been implemented in EV research. Adding case studies or examples of studies that adhered to MISEV standards would help illustrate their impact more vividly.
  2. Emphasis on Future Challenges: The manuscript briefly touches upon the future direction of the MISEV guidelines. However, a more in-depth discussion of the upcoming challenges and potential improvements for future versions would strengthen the impact of the paper. Specifically, it would be valuable to expand on how the EV community could tackle these challenges and the possible roles of emerging technologies.
  3. Standardization Gaps: The manuscript could be improved by further highlighting specific gaps that still remain in the standardization process of EV research. While it is clear that MISEV guidelines have improved over the years, an explicit discussion of current limitations would help emphasize areas that need further work.

Suggestions for Improvement

  1. Incorporate Examples and Case Studies: Adding real-life case studies of how the MISEV guidelines have helped standardize research would provide a practical perspective for readers. This could include a brief analysis of a few studies that successfully implemented the 2018 guidelines and how their adherence improved reproducibility.
  2. Expand on Future Directions: Consider adding a more detailed section that speculates on the future evolution of the MISEV guidelines. This could include possible additions to the guidelines that address emerging technologies, such as advanced omics approaches or new EV isolation techniques.
  3. Explicit Discussion of Limitations: Including a discussion that explicitly addresses gaps or limitations of the current MISEV guidelines would be beneficial. This could help readers understand what aspects of EV research are still problematic and could help shape future efforts in the field.

Additional references to incorporate based on liposomal studies utilizing non-invasive techniques such as neutron scattering:

a.      https://pubs.acs.org/doi/abs/10.1021/acs.langmuir.9b01534

b.      https://www.sciencedirect.com/science/article/abs/pii/S104620232400046X

c.       https://www.melaonin-research.com/index.php/MR/article/view/185

This manuscript is subject to minor revisions only.

Author Response

Response to the comments from Reviewer #1:

Comments 1: Incorporate Examples and Case Studies: Adding real-life case studies of how the MISEV guidelines have helped standardize research would provide a practical perspective for readers. This could include a brief analysis of a few studies that successfully implemented the 2018 guidelines and how their adherence improved reproducibility.

Response:

Thank you for your insightful comments and valuable suggestions to enhance the quality of our manuscript. We have cited more articles in the revised manuscript to discuss.

Revised manuscript (Page 5, Line 132-152):

In the process of separating EVs, it is often accompanied by other components. For example, LDL is more similar to exosomes due to its shape and size, density gradient centrifugation, size exclusion chromatography (SEC), ultrafiltration (UF) are more suitable to separate the two. And for soluble proteins, the use of immunocapture (IC) is more appropriate [13]. Of course, the combination of separation techniques is more helpful to achieve high purity samples, and both Density gradient ultracentrifugation (DGUC)-SEC and SEC-fast protein liquid chromatography (FPLC) combinations have significantly reduced EV/lipoprotein contamination [14, 15]. In addition to this, methods such as utilizing styrene-maleic acid [16], Simoa assay for ApoB-100 [17], and magnetic bead reagents [18] can be used to improve extracellular vesicle purification. For EV separation from cell culture media and blood, samples can be separated by combining SEC with UF to improve purity and yield [19, 20], and even a three-step protocol of polyethylene glycol (PEG) precipitation, gradient centrifugation, and SEC can be utilized for efficient removal of impurities and recovery of EVs [21]. Automated separation and hierarchical screening of size-based subpopulations have also been applied in the online immunoaffinity chromatography-asymmetric flow field flow separation (IAC-AF4) technique [22]. In conclusion, the selection of suitable separation strategies is crucial for different sample types, which will significantly affect the efficiency of EV extraction and the success of subsequent analysis. With the advancement of technology, emerging methods and combinations continue to emerge, providing more possibilities for efficient EV purification.

Comments 2: Expand on Future Directions: Consider adding a more detailed section that speculates on the future evolution of the MISEV guidelines. This could include possible additions to the guidelines that address emerging technologies, such as advanced omics approaches or new EV isolation techniques.

Response:

Thank you for this suggestion. We think this is an excellent suggestion. We have added more detailed section that speculates on the future evolution of the MISEV guidelines in the revised manuscript.

Revised manuscript (Page 10, Line 293-312):

In the current thriving landscape of machine learning, the EV field is also increasingly adopting machine learning techniques. These techniques are being used for EV recognition, classification, and component detection, greatly advancing the progress of disease diagnosis and prediction. For example, machine learning was used to identify, classify, and quantify EVs. the purity and inflammatory state of EVs were assessed through unsupervised machine learning [48]. This unsupervised learning method is able to automatically analyze data structures without pre-labeling, helping researchers better understand how EVs behave in different physiological and pathological states. Machine learning can also be used to analyze the components of EVs such as proteins, RNAs. through the combination of deep learning and TIRF, miRNAs of EVs were detected while their cancer origin was analyzed and diagnosed [49]. Not only that, the results of EV proteomics analysis with the help of machine learning are used to predict tumor invasion and proliferation ability and identify EVs of cancer origin. which is of great significance for revealing disease signaling pathways and biological processes and identifying potential drug targets [50-53]. Overall, the introduction of machine learning has brought new opportunities for EV research. Its application not only improves the efficiency and accuracy of experiments, but also provides new perspectives for clinical practice and promotes the development of precision medicine. In the future, whether the next version of MISEV will incorporate recommendations on machine learning is indeed a topic worthy of exploring in depth.

Comments 3: Explicit Discussion of Limitations: Including a discussion that explicitly addresses gaps or limitations of the current MISEV guidelines would be beneficial. This could help readers understand what aspects of EV research are still problematic and could help shape future efforts in the field.

Response:

Thank you for your valuable suggestion. We appreciate your recommendation to include a discussion that explicitly addresses the gaps or limitations of the current MISEV guidelines. We have added relevant discussion in the revised manuscript.

Revised manuscript (Page 3-4, Line 86-105, Page 9-10, Line 278-292):

Depending on the source of the sample (e.g., cell culture medium (CCM), blood, urine, etc.), there are significant differences in the methods used to collect samples and to isolate and store EVs. Therefore, a detailed documentation of the entire experimental conditions and procedures is essential to ensure the reproducibility of the experiments and the reliability of the results. For CCM, the report should include, but not be limited to, the composition and preparation of the medium, characteristics of EV-producing cells, culture conditions, frequency and method of cell culture medium harvesting, CCM storage, and whether serum, platelet lysate, or other complex additives were used. For blood samples, it is necessary to record donor characteristics, comply with blood collection requirements, avoid/remove platelet activation, and report that EV is enriched in blood samples. With the support of MIblood-EV, researchers can significantly improve the quality and reproducibility of studies involving blood EVs [9]. During urine processing, care needs to be taken to use cell-free urine and to document the method and results of the EV isolation considered, and the data on urinary EV and non-EV urine parameters should be collected. For more details, you can refer to the guidelines for urine EV [10, 11]. When isolating and storing EVs, it is important to regulate environmental conditions (e.g., pH and temperature) to maintain their functionality. Therefore, comprehensive and detailed experimental records not only help data analysis, but also provide the basis and reference for subsequent studies.

MISEV has been updated not only in terms of technology, but also in terms of researchers' deeper understanding of EV characteristics and applications. In the future, these innovations and collaborations will drive the development of more reproducible, transparent, and rigorous experimental methods, making the process of studying EV easier and more efficient, and thus providing higher quality studies for both basic research and clinical applications. Although MISEV2023 provides important guidance for EV research, it still faces some limitations and challenges. The lack of transparency of the experimental process makes it difficult to repeat experiments, which is also highlighted by MISEV2023. Second, EV isolation techniques in complex samples need to be improved, and there is a lack of efficient and simple isolation techniques. And some techniques still lack uniform standards. The functional mechanisms of EVs are still unclear, so there is still a long way to go to apply EVs to clinical use. In addition, it is worth noting that MISEV2023 lacks the categorization to distinguish some terms of cell membrane-derived vesicles [44, 45] and cell membrane-bound vesicles [46, 47], which are similar to EV characteristics, to avoid conceptual confusion among researchers.

Reviewer 2 Report

Comments and Suggestions for Authors

Minimal information for studies of extracellular vesicles (MISEV) gives EV researchers with great guidance and information about available techniques and the associated advantages and limitations. Zhang et al. reviewed the development of MISEV and the distinct changes in the MISEV. The minireview manuscript could be very interesting to EV researchers and researchers who wants to start EV research but with limited information of EVs. I only have two minor recommendations: 1)the goal of MISEV is not clearly demonstrated in the abstract and introduction so it is highly recommended to make it clear that reviewing and understanding the progress of MISEV is very important to improve the rigor, reproducibility and transparency in EV research. The authors can refer to this paper: 10.1038/s41417-024-00759-7. 2) There are not sufficient references to support the discussion and review of MISEV. The authors needs to add more references.

Author Response

Response to the comments from Reviewer #2:

Comments 1: Minimal information for studies of extracellular vesicles (MISEV) gives EV researchers with great guidance and information about available techniques and the associated advantages and limitations. Zhang et al. reviewed the development of MISEV and the distinct changes in the MISEV. The minireview manuscript could be very interesting to EV researchers and researchers who wants to start EV research but with limited information of EVs. I only have two minor recommendations: 1)the goal of MISEV is not clearly demonstrated in the abstract and introduction so it is highly recommended to make it clear that reviewing and understanding the progress of MISEV is very important to improve the rigor, reproducibility and transparency in EV research. The authors can refer to this paper: 10.1038/s41417-024-00759-7. 2) There are not sufficient references to support the discussion and review of MISEV. The authors needs to add more references

Response:

Thank you for your comments and valuable suggestions to improve the quality of our manuscript. We have added relevant content in the Abstract and Introduction (Brief history of EVs, ISEV, and MISEV) and added recommended articles in the revised manuscript.

Revised manuscript (Page 1-2, Line 10-17, 41-47):

In the tenth year since the first edition of MISEV was released in 2014, MISEV2023 has been reported in 2024 with the aim of refining the standard and improving the rigor, reproducibility, and transparency of extracellular vesicle (EV) research to clarify the requirements for experimental design of EVs, emphasize the importance of reproducible experimental results as well as encouraging openness of experimental information. The release of MISEV has significantly contributed to the quality of research in the field of EVs, which creates a more reliable research environment. However, despite the important role of MISEV, there is still a need for the EV researchers to continue to push for the widespread implementation of the guidelines to meet the evolving nature and challenges of EV research.

By providing clear definitions and classifications, detailed guidelines for experimental design, data reporting requirements, and standardization of sample treatment, MISEV2023 ensures the reliability and reproducibility of EV research. MISEV2023 encourages the sharing of experimental information and promotes the construction of databases to enhance the openness and transparency of studies. This guideline builds a solid foundation for basic research in the field of EV.

Reviewer 3 Report

Comments and Suggestions for Authors

Overall this a simple and reasonable summary of the MISEV progression which may be of use to authors trying to navigate the guidance. I have two small suggestions:

- replace use of "we", and stick to an impersonal summary.

- I did not understand why some text was blue/red in the tables. This needs clarifying.

Comments on the Quality of English Language

fine, just needs to avoid personal terms like "we".

Author Response

Response to the comments from Reviewer #3:

Comments 1:

Overall this a simple and reasonable summary of the MISEV progression which may be of use to authors trying to navigate the guidance. I have two small suggestions:

- replace use of "we", and stick to an impersonal summary.

- I did not understand why some text was blue/red in the tables. This needs clarifying.

Response:

Thank you very much for taking the time to review our manuscript. Your constructive comments and suggestions have been invaluable in improving the quality of our work. We have deleted “we” and improved the context of the manuscript. With regard to the blue font in the table, we have explained this at the bottom of the tables and removed the red font and changed it to bold for emphasis.

Reviewer 4 Report

Comments and Suggestions for Authors

In the manuscript “Minimal information for studies of extracellular vesicles 2 (MISEV): Ten-year evolution (2014-2023)” The authors review the evolution of the field of extracellular vesicle research, as it pertains to the MISEV guidelines published over the last 10 years. The manuscript is generally well written and easy to read. The tables give very clear points on the differences between each edition and divided in easy-to-understand points. The article is a nice recap and representation for those in the field already and would be a beneficial guide to beginners starting out in this complex field. I do suggest some modifications to improve the review:

1.      Some more information could be added to the technique sections. Such as the proteins eligible for characterisation, such as the gold standard targets for the surface markers and the negative controls. To say WB with a minimum number of targets is helpful, but to name those eligible targets one can ‘choose’ from, will give an extra level. This may pertain to other techniques where relevant. 

2.      In the sections without tables, it feels like there is less detailed information presented. It would be nice to either give more detail to cover what would be in the table in as clear and detailed way, or add in tables in these sections to provide it. 

3.      While the purpose is to analyse the MISEV guidelines, it may also be interesting, as an extension of the evolution of the field, to mention some of the new technologies coming out that provide MISEV-abiding analysis of EVs in a new modality. To show that technology has also evolved to meet the field. 

4.      The last section with the summary and discussion, specifically the first paragraph, should be re-written. There are a few language errors but also the use of metaphors and ‘emotion’ related terms (specifically: love) feel out of context given the tone of the paper preceding the section. Of course, it is important to outline the importance of MISEV, however this could be portrayed in a more professional manner.

Overall, the paper gives nice insights to the precise evolution of the field.

Author Response

Response to the comments from Reviewer #4:

Comments 1:

Some more information could be added to the technique sections. Such as the proteins eligible for characterization, such as the gold standard targets for the surface markers and the negative controls. To say WB with a minimum number of targets is helpful, but to name those eligible targets one can ‘choose’ from, will give an extra level. This may pertain to other techniques where relevant.

Response:

Thank you for reviewing our article and for your valuable comments. We value your feedback and have revised and added to the article accordingly.

Revised manuscript (Page 6, Line 179-182):

Proteins like CD9, CD63, CD81, TSG101, ALIX, and GAPDH can be used to assess the presence of EVs, but it's worth noting that not all EVs display these proteins. Negative protein markers can be chosen from proteins such as Calnexin, Histones or GM130. Notably, in addition to testing for EV marker proteins, the presence of NVEP in EV preparations needs to be evaluated to ensure sample purity and specificity.

Comments 2:

In the sections without tables, it feels like there is less detailed information presented. It would be nice to either give more detail to cover what would be in the table in as clear and detailed way, or add in tables in these sections to provide it.

Response:

We feel great thanks for your professional review work on our article. The sections of EV release and uptake, Functional studies, and EV analysis in vivo have been supplemented with additional information.

Revised manuscript (Page 8-9, Line 230-238, 246-251, 258-261):

Learning about the uptake and release mechanisms of EVs can deepen our understanding of intercellular communication and provide new perspectives on the application of EVs as drug carriers. EVs possess the ability to target specific cells, good biocompatibility, and excellent drug-carrying capacity, which enables them to efficiently encapsulate small-molecule drugs, RNAs, and proteins. In addition, EVs are able to penetrate cell membranes and release drugs directly into target cells, while protecting them from degradation, thus improving bioavailability [37-40]. These properties have enabled EVs to show a wide range of potential applications in therapeutic areas such as cancer, cardiovascular diseases, autoimmune and neurological diseases [41, 42].

When conducting functional studies, dose-response and time-course assays are recommended for a comprehensive assessment of the biological effects of EVs. Appropriate negative EV controls should be used to ensure the accuracy of the experiments. In addition, non-EV negative controls need to be evaluated to exclude possible background effects. This will help improve the reliability and relevance of the findings, leading to a better understanding of EV functions and mechanisms.

When performing in vivo experiments, detailed reporting of labeling, detection techniques and exogenous EV administration parameters is required to facilitate replication studies. Also, attention is paid to the effects of EV labeling on biodistribution and function, as well as behavioral differences between endogenous and exogenous EVs.

Comments 3:

While the purpose is to analyze the MISEV guidelines, it may also be interesting, as an extension of the evolution of the field, to mention some of the new technologies coming out that provide MISEV-abiding analysis of EVs in a new modality. To show that technology has also evolved to meet the field.

Response:

Thank you for your positive comments and valuable suggestions to improve the quality of our manuscript. In the manuscript we have added novel combinations of techniques and characterization methods not mentioned in MISEV2023.

Revised manuscript (Page 7, Line 199-216):

New advances in high-resolution microscopy techniques, in conjunction with innovative labeling strategies, have made it possible to explore kinetics and pharmacogenetics at the nanoscale [26-29]. The application of total internal reflection fluorescence microscopy (TIRFM) with dynamic correlative light and electron microscopy (CLEM) provides a powerful tool for observing the fusion of EVs with the cytoplasmic membrane and real-time in vivo imaging of EVs [30-32]. By utilizing TIRFM visualization for automatic image acquisition and quantification to design a tunable micropattern-array assay detection method in realizing EV sorting while detecting its RNA and proteins in situ, it simplifies the operation process and improves the measurement accuracy [33]. Single particle interferometric reflectance imaging sensing (SP-IRIS) technology in combination with single-molecule fluorescent in situ hybridization (smFISH) can likewise detect the RNA and surface proteins of EVs. In terms of characterizing the size of EVs, in addition to the widely used nanoparticle tracking analysis (NTA), dynamic light scattering (DLS), and resistive pulse sensing (RPS), nuclear magnetic resonance (NMR) has gradually emerged as an effective means that can be used to measure the size of extracellular vesicles [34]. Limited by factors such as EV heterogeneity and yield, studies have begun to use only flow cytometry instead of WB or both flow cytometry and WB techniques as a means of detecting EV markers [35, 36].

Comments 4:

The last section with the summary and discussion, specifically the first paragraph, should be re-written. There are a few language errors but also the use of metaphors and ‘emotion’ related terms (specifically: love) feel out of context given the tone of the paper preceding the section. Of course, it is important to outline the importance of MISEV, however this could be portrayed in a more professional manner.

Response:

Thank you for your constructive feedback. We sincerely apologize for any shortcomings in our previous manuscript and assure you that we have taken great care to revise and polish it thoroughly. We have made concerted efforts to enhance the clarity, accuracy, and coherence of our writing.
